# ENSEMBLE ROBUSTNESS AND GENERALIZATION OF STOCHASTIC DEEP LEARNING ALGORITHMS

**Tom Zahavy[‡], Alexander Sivak[∗∗], Bingyi Kang[∗], Jiashi Feng[∗], Huan Xu[†] & Shie Mannor[‡]**

‡ Department of EE, Technion
∗∗ Department of CS, Technion
† Department of ISE, National University of Singapore
∗ Department of ECE, National University of Singapore
`{tomzahavy@campus,silex@cs,shie@ee}.technion.ac.il`
`{bingykang,jshfeng}@gmail.com`
`huan.xu@isye.gatech.edu`

## ABSTRACT

The question why deep learning algorithms generalize so well has attracted increasing research interest. However, most of the well-established approaches, such as hypothesis capacity, stability or sparseness, have not provided complete explanations (Zhang et al., 2017; Kawaguchi et al., 2017). In this work, we focus on the robustness approach (Xu & Mannor, 2012), i.e., if the error of a hypothesis will not change much due to perturbations of its training examples, then it will also generalize well. As most deep learning algorithms are stochastic (e.g., Stochastic Gradient Descent, Dropout, and Bayes-by-backprop), we revisit the robustness arguments of Xu & Mannor, and introduce a new approach – ensemble robustness – that concerns the robustness of a *population* of hypotheses. Through the lens of ensemble robustness, we reveal that a stochastic learning algorithm can generalize well as long as its sensitiveness to adversarial perturbations is bounded in average over training examples. Moreover, an algorithm may be sensitive to some adversarial examples (Goodfellow et al., 2015) but still generalize well. To support our claims, we provide extensive simulations for different deep learning algorithms and different network architectures exhibiting a strong correlation between ensemble robustness and the ability to generalize.

## 1 INTRODUCTION

Deep Neural Networks (DNNs) have been successfully applied in many artificial intelligence tasks, providing state-of-the-art performance and a remarkably small generalization error. On the other hand, DNNs often have far more trainable model parameters than the number of samples they are trained on and were shown to have a large enough capacity to memorize the training data (Zhang et al., 2017). The findings of Zhang et al. suggest that classical explanations for generalization cannot be applied directly to DNNs and motivated researchers to look for new complexity measures and explanations for the generalization deep neural networks (Bartlett et al., 2017; Neyshabur et al., 2017; Arpit et al., 2017; Kawaguchi et al., 2017). However, in this work, we focus on a different approach to study generalization of DNNs, i.e., the connection between the robustness of a deep learning algorithm and its generalization performance. Xu & Mannor have shown that if an algorithm is robust (*i.e.*, its empirical loss does not change dramatically for perturbed samples), its generalization performance can also be guaranteed. However, in the context of DNNs, practitioners observe **contradicting** evidence between these two attributes. On the one hand, DNNs generalize well, and on the other, they are fragile to adversarial perturbation on the inputs (Szegedy et al., 2014; Goodfellow et al., 2015). Nevertheless, algorithms that try to improve the robustness of learning algorithms have been shown to improve the generalization of deep neural networks. Two examples are adversarial training, i.e., generating adversarial examples and training on them (Szegedy et al., 2014; Goodfellow et al., 2015; Shaham et al., 2015), and Parseval regularization (Cisse et al., 2017), i.e., minimizing the Lifshitz constant of the network to guarantee low robustness. While these meth-

ods minimize the robustness implicitly, their empirical success Indicates a connection between the robustness of an algorithm and its ability to generalize.

To solve this contradiction, we revisit the robustness argument in (Xu & Mannor, 2012) and present *ensemble robustness*, to characterize the generalization performance of deep learning algorithms. Our proposed approach is not intended to give tight bounds for general deep learning algorithms, but rather to pave the way for addressing the question: how can deep learning perform so well while being fragile to adversarial examples? Answering this question is difficult, yet we present evidence in both theory and simulation suggesting that *ensemble robustness* explains the generalization performance of deep learning algorithms.

Ensemble robustness concerns the fact that a randomized algorithm (*e.g.*, Stochastic Gradient Descent (SGD), Dropout (Srivastava et al., 2014), Bayes-by-backprop (Blundell et al., 2015), etc.) produces a distribution of hypotheses instead of a deterministic one. Therefore, ensemble robustness takes into consideration robustness of the *population* of the hypotheses: even though some hypotheses may be sensitive to perturbation on inputs, an algorithm can still generalize well as long as most of the hypotheses sampled from the distribution are robust on average. Kawaguchi et al. (2017) took a different approach and claimed that deep neural networks could generalize well despite nonrobustness. However, our definition of ensemble robustness together with our empirical findings suggest that deep learning methods are typically robust although being fragile to adversarial examples.

Through ensemble robustness, we prove that the following holds with a high probability: randomized learning algorithms can generalize well as long as its output hypothesis has bounded sensitiveness to perturbation in average (see Theorem 1). Specified for deep learning algorithms, we reveal that if hypotheses from different runs of a deep learning method perform consistently well in terms of robustness, the performance of such deep learning method can be confidently expected. Moreover, each hypothesis may be sensitive to some adversarial examples as long as it is robust on average.

Although ensemble robustness may be difficult to compute analytically, we demonstrate an empirical estimate of ensemble robustness and investigate the role of ensemble robustness via extensive simulations. The results provide supporting evidence for our claim: ensemble robustness consistently explains the generalization performance of deep neural networks. Furthermore, ensemble robustness is measured solely on training data, potentially allowing one to use the testing examples for training and selecting the best model based on its ensemble robustness.

## 2 RELATED WORKS

Xu *et al.*( 2012) proposed to consider model robustness for estimating generalization performance for deterministic algorithms, such as for SVM (Xu et al., 2009b) and Lasso (Xu et al., 2009a). They suggest using robust optimization to construct learning algorithms, *i.e.*, minimizing the empirical loss concerning the adversarial perturbed training examples.

Introducing stochasticity to deep learning algorithms has achieved great success in practice and also receives theoretical investigation. Hardt *et al.* (2015) analyzed the stability property of SGD methods, and Dropout (Srivastava et al., 2014) was introduced as a way to control over-fitting by randomly omitting subsets of features at each iteration of a training procedure. Different explanations for the empirical success of dropout have been proposed, including, avoiding over-fitting as a regularization method (Baldi & Sadowski, 2013; Wager et al., 2013; Jain et al., 2015) and explaining dropout as a Bayesian approximation for a Gaussian process (Gal & Ghahramani, 2015). Different from those works, this work will extend the results in (Xu & Mannor, 2012) to randomized algorithms, to analyze them from an ensemble robustness perspective.

Robustness and ensemble robustness share some similarities with stability, a related yet different property of learning algorithms that also guarantees generalization. An algorithm is stable if it produces an output hypothesis that is not sensitive to the sampling of the empirical data set. In more detail, if a training example is replaced with another example from the same distribution, the training error will not change much; see (Bousquet & Elisseeff, 2002) for more details, and (Elisseeff et al., 2005) for a discussion on randomized algorithms. We emphasize that robustness and stability are different properties; robustness concerns global modifications of the training data while stability is more local in that sense. Moreover, robustness concerns attributes of a single hypothesis, while

stability concerns two (one for the original data set and one for the modified one). Finally, a learning algorithm may be both stable and robust, e.g., SVM (Xu et al., 2009b), or robust but not stable, e.g., Lasso Regression (Xu et al., 2009a).

Adversarial examples for deep neural networks were first introduced in (Szegedy et al., 2014), while some recent works propose to utilize them as a regularization technique for training deep models (Goodfellow et al., 2015; Gu & Rigazio, 2014; Shaham et al., 2015). However, all of those works attempt to find the "worst case" examples in a local neighborhood of the original training data and are not focused on measuring the global robustness of an algorithm nor on studying the connection between robustness and generalization.

## 3   PRELIMINARIES

In this work, we investigate the generalization property of stochastic learning algorithms in deep neural networks, by establishing their PAC bounds. In this section, we provide some preliminary facts that are necessary for developing the approach of ensemble robustness. After introducing the problem setup we are interested in, we in particular highlight the inherent randomness of deep learning algorithms and give a formal description of randomized learning algorithms. Then, we briefly review the relationship between robustness and generalization performance established in (Xu & Mannor, 2012).

**Problem setup**   We now introduce the learning setup for deep neural networks, which follows a standard one for supervised learning. More concretely, we have $\mathcal{Z}$ and $\mathcal{H}$ as the sample set and the hypothesis set respectively. The training sample set $\mathbf{s} = \{s_1, \ldots, s_n\}$ consists of $n$ i.i.d. samples generated by an unknown distribution $\mu$, and the target of learning is to obtain a neural network that minimizes *expected* classification error over the i.i.d. samples from $\mu$. Throughout the paper, we consider the training set $\mathbf{s}$ with a fixed size of $n$.

We denote the learning algorithm as $\mathcal{A}$, which is a mapping from $\mathcal{Z}^n$ to $\mathcal{H}$. We use $\mathcal{A} : \mathbf{s} \to h_\mathbf{s}$ to denote the learned hypothesis given the training set $\mathbf{s}$. We consider the loss function $\ell(h, z)$ whose value is nonnegative and upper bounded by $M$. Let $\mathcal{L}(\cdot)$ and $\ell_{\mathrm{emp}}(\cdot)$ denote the expected error and the training error for a learned hypothesis $h_\mathbf{s}$, *i.e.*,

$$\mathcal{L}(h_\mathbf{s}) \triangleq \mathbb{E}_{z\sim\mu}\ell(h_\mathbf{s}, z), \text{ and } \ell_{\mathrm{emp}}(h_\mathbf{s}) \triangleq \frac{1}{n}\sum_{s_i \in \mathbf{s}} \ell(h_\mathbf{s}, s_i). \tag{1}$$

We are going to characterize the generalization error $|\mathcal{L}(h_\mathbf{s}) - \ell_{\mathrm{emp}}(h_\mathbf{s})|$ of deep learning algorithms in the following section.

**Randomized algorithms**   Most of modern deep learning algorithms are in essence randomized ones, which map a training set $\mathbf{s}$ to a *distribution* of hypotheses $\Delta(\mathcal{H})$ instead of a single hypothesis. For example, running a deep learning algorithm $\mathcal{A}$ with dropout for multiple times will produce different hypotheses which can be deemed as samples from the distribution $\Delta(\mathcal{H})$. Therefore, before proceeding to analyze the performance of deep learning, we provide a formal definition of randomized learning algorithms here.

**Definition 1** (Randomized Algorithms)**.** *A randomized learning algorithm $\mathcal{A}$ is a function from $\mathcal{Z}^n$ to a set of distributions of hypotheses $\Delta(\mathcal{H})$, which outputs a hypothesis $h_\mathbf{s} \sim \Delta(\mathcal{H})$ with a probability $\pi_\mathbf{s}(h)$.*

When learning with a randomized algorithm, the target is to minimize the expected empirical loss for a specific output hypothesis $h_\mathbf{s}$, similar to the ones in (1). Here $\ell$ is the loss incurred by a specific output hypothesis by one instantiation of the randomized algorithm $\mathcal{A}$.

Examples of the internal randomness of a deep learning algorithm $\mathcal{A}$ include dropout rate (the parameter for a Bernoulli distribution for randomly masking certain neurons), random shuffle among training samples in SGD, the initialization of weights for different layers, to name a few.

**Robustness and generalization**   Xu & Mannor (2012) established the relation between algorithmic robustness and generalization for the first time. An algorithm is robust if the following holds: if

two samples are close to each other, their associated losses are also close. For being self-contained, we here briefly review the algorithmic robustness and its induced generalization guarantee.

**Definition 2** (Robustness, Xu & Mannor (2012)). *Algorithm $\mathcal{A}$ is $(K, \epsilon(\cdot))$ robust, for $K \in \mathbb{N}$ and $\epsilon(\cdot) : \mathcal{Z}^n \to \mathbb{R}$, if $\mathcal{Z}$ can be partitioned into $K$ disjoint sets, denoted by $\{C_i\}_{i=1}^K$, such that the following holds for all $\mathbf{s} \in \mathcal{Z}^n$:*

$$\forall s \in \mathbf{s}, \forall z \in \mathcal{Z}, \forall i = 1, \ldots, K :$$
$$\text{if } s, z \in \mathcal{C}_i, \text{ then } |\ell(\mathcal{A}_{\mathbf{s}}, s) - \ell(\mathcal{A}_{\mathbf{s}}, z)| \leq \epsilon(n).$$

Based on the above robustness property of algorithms, Xu *et al.* (Xu & Mannor, 2012) prove that a robust algorithm also generalizes well.

Motivated by their results, Shaham *et al.* (Shaham et al., 2015) proposed adversarial training algorithm to minimize the empirical loss over synthesized adversarial examples. However, those results cannot be applied for characterizing the performance of modern deep learning models well.

## 4  ENSEMBLE ROBUSTNESS

To explain the proper performance of deep learning, one needs to understand the internal randomness of deep learning algorithms and the population performance of the multiple possible hypotheses. Intuitively, a single output hypothesis cannot be robust to adversarial perturbation on training samples and the deterministic robustness argument in (Xu & Mannor, 2012) cannot be applied here. Fortunately, deep learning algorithms output the hypothesis sampled from a distribution of hypotheses. Therefore, even if some samples are not "nice" for one specific hypothesis, they aren't likely to fail *most* of the hypothesis from the produced distribution. Thus, deep learning algorithms generalize well. Such intuition motivates us to introduce the concept of *ensemble robustness* that is defined over the distribution of output hypotheses of a deep learning algorithm.

**Definition 3** (Ensemble Robustness). *A randomized algorithm $\mathcal{A}$ is $(K, \bar{\epsilon}(n))$ ensemble robust, for $K \in \mathbb{N}$ and $\bar{\epsilon}(n)$, if $\mathcal{Z}$ can be partitioned into $K$ disjoint sets, denoted by $\{C_i\}_{i=1}^K$, such that the following holds for* all $\mathbf{s} \in \mathcal{Z}^n$:

$$\forall s \in \mathbf{s}, \forall i = 1, \ldots, K :$$
$$\text{if } s \in \mathcal{C}_i, \text{ then } \mathbb{E}_{\mathcal{A}} \max_{z \in \mathcal{C}_i} |\ell(\mathcal{A}_{\mathbf{s}}, s) - \ell(\mathcal{A}_{\mathbf{s}}, z)| \leq \bar{\epsilon}(n).$$

*Here the expectation is taken w.r.t. the internal randomness of the algorithm $\mathcal{A}$.*

An algorithm with strong ensemble robustness can provide good generalization performance in expectation w.r.t. the generated hypothesis, as stated in the following theorem. We note that the proofs for all the theorems that we present in this section can be found supplementary material. Also, the supplementary material holds an additional proof for the special case of Dropout.

**Theorem 1.** *Let $\mathcal{A}$ be a randomized algorithm with $(K, \bar{\epsilon}(n))$ ensemble robustness over the training set $\mathbf{s}$, with $|\mathbf{s}| = n$. Let $\Delta(\mathcal{H}) \leftarrow \mathcal{A} : \mathbf{s}$ denote the output hypothesis distribution of $\mathcal{A}$. Then for any $\delta > 0$, with probability at least $1 - \delta$ with respect to the random draw of the $\mathbf{s}$ and $h \sim \Delta(\mathcal{H})$, the following holds:*

$$|\mathcal{L}(h) - \ell_{\text{emp}}(h)| \leq \sqrt{\frac{nM\bar{\epsilon}(n) + 2M^2}{\delta n}}.$$

Note that in the above theorem, we hide the dependency of the generalization bound on $K$ in ensemble robustness measure $\bar{\epsilon}(n)$. Generally, there is a trade-off between $\bar{\epsilon}(n)$ and $K$, the larger K is, the smaller $\bar{\epsilon}(n)$ is due to the finer partition. This tradeoff is more evident in the bound of Theorem 2, see also (Xu & Mannor, 2012). Studying the asymptotics of $\bar{\epsilon}(n)$ is hard for general algorithms and deep networks, yet, it can be done for simpler learning algorithms. For example, for linear SVM, $\bar{\epsilon}(n)$ is equivalent to the covering number (Xu et al., 2009b).

Due to space limitations, all the technical lemmas and details of the proofs throughout the paper are deferred to supplementary material. Theorem 1 leads to the following corollary which gives a way to minimize expected loss directly.

**Corollary 1.** *Let $\mathcal{A}$ be a randomized algorithm with $(K, \bar{\epsilon}(n))$ ensemble robustness. Let $C_1, \ldots, C_K$ be a partition of $\mathcal{Z}$, and write $z_1 \sim z_2$ if $z_1, z_2$ fall into the same $C_k$. If the training sample $\mathbf{s}$ is generated by i.i.d. draws from $\mu$, then with probability at least $1 - \delta$, the following holds over $h \in \mathcal{H}$*

$$\mathcal{L}(h) \leq \frac{1}{n} \sum_{i=1}^{n} \max_{z_i \sim s_i} \ell(h, z_i) + \sqrt{\frac{nM\bar{\epsilon}(n) + 2M^2}{\delta n}}.$$

Corollary 1 suggests that one can minimize the expected error of a deep learning algorithm effectively through minimizing the empirical error over the training samples $s_i$ perturbed in an adversarial way. In fact, such an adversarial training strategy has been exploited in (Goodfellow et al., 2015; Shaham et al., 2015).

**Theorem 2.** *Let $\mathcal{A}$ be a randomized algorithm with $(K, \bar{\epsilon}(n))$ ensemble robustness over the training set $\mathbf{s}$, where $|\mathbf{s}| = n$. Let $\Delta(\mathcal{H})$ denote the output hypothesis distribution of the algorithm $\mathcal{A}$ on the training set $\mathbf{s}$. Suppose following variance bound holds:*

$$\mathrm{var}_{\mathcal{A}} \left[ \max_{z \sim s_i} |\ell(\mathcal{A}_{\mathbf{s}}, s_i) - \ell(\mathcal{A}_{\mathbf{s}}, z)| \right] \leq \alpha$$

*Then for any $\delta > 0$, with probability at least $1 - \delta$ with respect to the random draw of the $\mathbf{s}$ and $h \sim \Delta(\mathcal{H})$, we have*

$$|\mathcal{L}(\mathcal{A}_{\mathbf{s}}) - \ell_{\mathrm{emp}}(\mathcal{A}_{\mathbf{s}})| \leq \bar{\epsilon}(n) + \frac{1}{\sqrt{2\delta}}\alpha + M\sqrt{\frac{2K \ln 2 + 2\ln(1/\delta)}{n}}$$

Theorem 2 implies that Ensemble robustness is a "weaker" requirement for the model compared with Robustness proposed in (Xu & Mannor, 2012). To see this, consider the trade-off between the expectation and variance of ensemble robustness on two extreme examples. When $\alpha = 0$, we do not allow any variance in the output of the algorithm $\mathcal{A}$. Thus, $\mathcal{A}$ reduces to a deterministic one. To achieve the above upper bound, it is required that the output hypothesis satisfies $\max_{z \in C_i} |\ell(h, s_i) - \ell(h, z)| \leq \epsilon(n)$. However, due to the intriguing property of deep neural networks (Szegedy et al., 2014), the deterministic model robustness measure $\epsilon(n)$ (ref. Definition 2) is usually large. In contrast, when the hypotheses variance $\alpha$ can be large enough, there are multiple possible output hypotheses from the distribution $\Delta(\mathcal{H})$. We fix the partition of $\mathcal{Z}$ as $C_1, \ldots, C_K$. Then,

$$\begin{aligned}
&\mathbb{E}_{\mathcal{A}}[\max_{z \sim s \in \mathbf{s} \cap C_i} |\ell(h, s) - \ell(h, z)|] \\
&= \sum_{j \in \Delta(\mathcal{H})} \mathbb{P}\{h = h_j\} \max_{z \sim s \in \mathbf{s} \cap C_i} |\ell(h_j, s) - \ell(h_j, z)| \\
&\leq \sum_{j \in \Delta(\mathcal{H})} \mathbb{P}\{h = h_j\} \max_{z \sim s \in \mathbf{s} \cap C_i} \max_{h \in \Delta\mathcal{H}} |\ell(h, s) - \ell(h, z)| \\
&\leq \max_{z \sim s \in \mathbf{s} \cap C_i} \max_{h \in \Delta(\mathcal{H})} |\ell(h, s) - \ell(h, z)|.
\end{aligned}$$

Therefore, allowing certain variance on produced hypotheses, a randomized algorithm can tolerate the non-robustness of some hypotheses to certain samples. As long as the ensemble robustness is small, the algorithm can still perform well. Indeed, in the following section we demonstrate through simulations that generalization of deep learning models is more correlated with ensemble robustness than robustness.

## 5 SIMULATIONS

This section is devoted to simulations for quantitatively and qualitatively demonstrating how ensemble robustness of a deep learning method explains its performance. We first introduce our experiment settings and implementation details.

### 5.1 EXPERIMENT SETTINGS

**Data sets** We conduct simulations on two benchmarks. MNIST, a dataset of handwritten digit images (28x28) with $50,000$ training samples and $10,000$ test samples (LeCun et al., 1998). NotM-NIST[1], a "mnist like database" containing font glyphs for the letters A through J (10 classes). The training set contains $367,440$ samples and $18,724$ testing examples. The images (for both data sets) were scaled such that each pixel is in the range $[0, 1]$. We note that we did not use the cross-validation data.

**Network architecture and parameter setting** Without explicit explanation, we use multi-layer perceptrons throughout the simulations. All networks we examined are composed of three fully connected layers, each of which is followed by a rectified linear unit on top. The output of the last fully-connected layer is fed to a 10-way softmax. To avoid the bias brought by specific network architecture on our observations, we sample at random the number of units in each layer (uniformly over $\{400, 800, 1200\}$ units) and the learning rate (uniformly over $[0.005, 0.05]$ for SGD, and uniformly over $[0.05, 0.5]$ for Bayes-by-backprop). Finally, we used a mini-batch of $128$ training examples at a time.

**Compared algorithms** We evaluate and compare ensemble robustness as well as the generalization performance for following $4$ deep learning algorithms. (1) **Explicit ensembles**, i.e., using a stochastic algorithm to train different members of the ensemble by running the algorithm multiple times with different seeds. In practice, this was implemented using SGD as the stochastic algorithm, trained to minimize the cross-entropy loss. (2) **Implicit ensembles**, i.e., learning a probability distribution on the weights of a neural network and sampling ensemble members from it. This was implemented with the *Bayes-by-backprop* (Blundell et al., 2015) algorithm, a recent approach for training Bayesian Neural Networks. It uses backpropagation to learn a probability distribution on the weights of a neural network by minimizing the expected lower bound on the marginal likelihood (or the variational free energy). Methods 3 and 4 correspond for adding adversarial training (Szegedy et al., 2014; Goodfellow et al., 2015; Shaham et al., 2015) to the ensemble methods, where the magnitude of perturbation is measured by its $\ell_2$ norm and is sampled uniformly over $\{0.1, 0.3, 0.5\}$ to avoid sampling bias. From now on, a specific configuration will refer to a unique set of these parameters (algorithm type, network width, learning rate and perturbation norm).

### 5.2 EMPIRICAL ENSEMBLE ROBUSTNESS AND GENERALIZATION

We now present simulations that empirically validate Theorem 1, *i.e.*, that the ensemble robustness of a DNN (measured on the training set) is highly correlated with its generalization performance. But empirically evaluating ensemble robustness of deep neural nets is hard for two reasons.

First, it is not clear how to define the partition of the samples into sets $\{C_i\}_{i=1}^K$. To deal with this challenge, we use adversarial examples, and define $K = n$ partitions such implicitly, such that each partition contains a small $\ell_2$ ball around each training example. We then approximate the loss change in each partition using the adversarial example, i.e., approximating the maximal loss in the partition using the adversarial example. While this approximation is loose, we will soon show that empirically, it is correlated with generalization. We emphasize that under this partition, there is no violation of the i.i.d assumption for general stochastic algorithms, but it is violated in the case of adversarial training (since the adversarial examples used for training are not sampled i.i.d). Despite the latter observation, we measured even stronger correlation for these algorithms.

Second, ensemble robustness involves taking an expectation over all the possible output hypothesis. Hence it is computationally intractable to measure ensemble robustness for deep learning algorithms exactly. In this simulation, we take the empirical average of robustness to adversarial perturbation from $5$ different hypotheses of the same learning algorithm as its ensemble robustness. In the case of the SGD variants, for each configuration, we collect an ensemble of output hypotheses by repeating the training procedures using the same configuration while using different random seeds. In the case of the Bayes-by-backprop methods, the algorithm explicitly outputs a distribution over output hypothesis, so we simply sample the networks from the learned weight distribution.

---

[1] http://yaroslavvb.blogspot.com/2011/09/notmnist-dataset.html

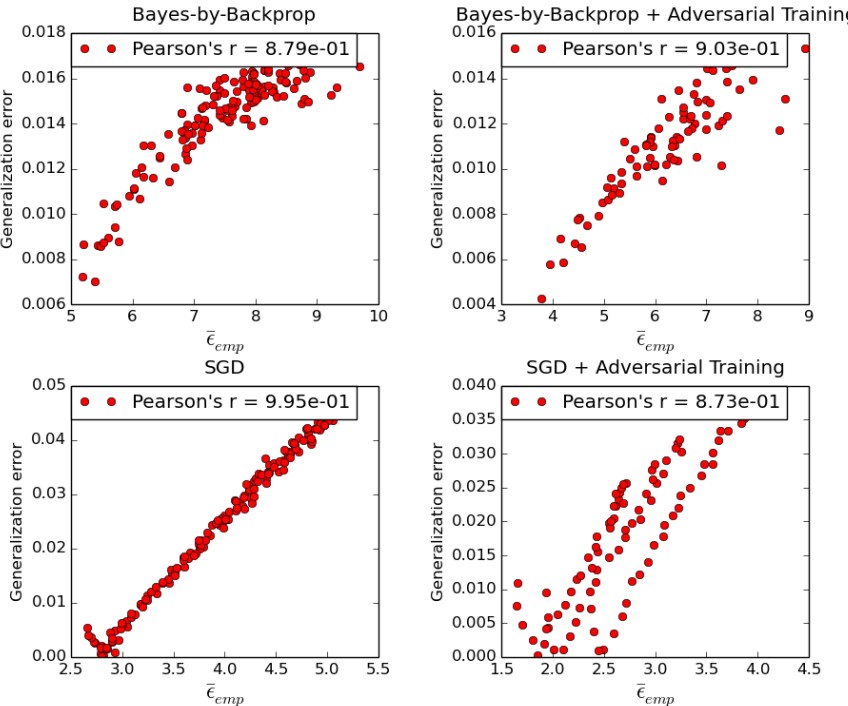

Figure 1: Results for **MNIST**. Empirical ensemble robustness $\bar{\epsilon}_{\mathrm{emp}}$ (x-axis) vs generalization error (y-axis). Results are given for four different deep learning algorithms.

In particular, we aim to empirically demonstrate that a deep learning algorithm with stronger ensemble robustness presents better generalization performance (Theorem 1). Recall the definition of ensemble robustness in Definition 3, another obstacle in calculating ensemble robustness is to find the most adversarial perturbation $\Delta s$ (or equivalently the most adversarial example $z = s + \Delta s$) for a specific training sample $s \in \mathbf{s}$ within a partition set $C_i$. We therefore employ an approximate search strategy for finding the adversarial examples. More concretely, we optimize the following first-order Taylor expansion of the loss function as a surrogate for finding the adversarial example:

$$\Delta s_i \in \underset{\|\Delta s_i\| \leq r}{\arg \max} \quad \ell(s_i) + \langle \nabla \ell_{s_i}(s), \Delta s_i \rangle, \tag{2}$$

with a pre-defined magnitude constraint $r$ on the perturbation $\Delta s_i$. In the simulations, we vary the magnitude $r$ in order to calculate the empirical ensemble robustness at different perturbation levels.

We then calculate the *empirical* ensemble robustness by averaging the difference between the loss of the algorithm on the training samples and the adversarial samples output by the method in (2):

$$\bar{\epsilon}_{\mathrm{emp}} = \frac{1}{T} \sum_{t=1}^{T} \max_{i \in \{1, \ldots, n\}} |\ell(\mathcal{A}_{\mathbf{s}}^{(t)}, s_i) - \ell(\mathcal{A}_{\mathbf{s}}^{(t)}, s_i + \Delta s_i)|, \tag{3}$$

with $T = 10$ denoting the size of the ensemble.

We emphasize that $\bar{\epsilon}(n)$ (Theorem 1) and the empirical approximation $\bar{\epsilon}(n)_{emp}$ measure the **non robustness** of an algorithm, *i.e.*, an algorithm is more robust if $\bar{\epsilon}(n)$ is smaller.

### 5.3 RESULTS

The generalization performance of different learning algorithms and different networks compared with the empirical ensemble robustness on MNIST is given in Figure 1. Notice that the x-axis corresponds to the *empirical* ensemble robustness (Equation 3), and the y-axis corresponds to the

test error. Examining Figure 1 we observe a high correlation between ensemble robustness and generalization for all learning algorithms, *i.e.*, algorithms that are more robust (have lower $\bar{\epsilon}(n)$) generalize better on this data set.

Figure 2 in the appendix presents similar results on the notMNIST dataset, although we observe lower (yet positive) correlation for the Bayes-by-backprop algorithm in this case. These observations support our claim on the relation between ensemble robustness and algorithm generalization performance in Theorem 1.

We also compare ensemble robustness with robustness on MNIST in Table 1, where robustness is measured similarly to ensemble robustness using Equation 3 but with $T = 1$ (while $T = 10$ for ensemble robustness). Indeed, we observe that averaging over instances of the same algorithm, exhibits a higher correlation between generalization and robustness, i.e., ensemble robustness is a better estimation of the generalization performance than standard robustness.

| Data set. | MNIST | |
|---|---|---|
| Metric | Robustness | Ensemble Robustness |
| SGD | 0.978 | 0.995 |
| SGD + adversarial training | 0.854 | 0.873 |
| Bayes-by-backprop | 0.759 | 0.879 |
| Bayes-by-backprop + adversarial training | 0.834 | 0.903 |

Table 1: Empirical robustness vs. ensemble robustness.

## 6 CONCLUSIONS

In this paper, we investigated the generalization ability of stochastic deep learning algorithm based on their ensemble robustness; *i.e.*, the property that if a testing sample is similar to a training sample, then its loss is close to the training error. We established both theoretically and experimentally evidence that ensemble robustness of an algorithm, measured on the training set, indicates its generalization performance well. Moreover, our theory and experiments suggest that DNNs may be robust (and generalize) while being fragile to specific adversarial examples. Measuring ensemble robustness of stochastic deep learning algorithms may be computationally prohibitive as one needs to sample several output hypotheses of the algorithm. Thus, we demonstrated that by learning the probability distribution of the weights of a neural network explicitly, e.g., via variational methods such as Bayes-by-backprop, we can still observe a positive correlation between robustness and generalization while using fewer computations, making ensemble robustness feasible to measure.

As a direct consequence, one can potentially measure the generalization error of an algorithm without using testing examples. In future work, we plan to further investigate if ensemble robustness can be used for model selection instead of cross-validation (and hence, increasing the training set size), in particular in problems that have a small training set. A different direction is to study the resilience of deep learning methods to adversarial attacks (Papernot et al., 2016). Strauss et al. (2017) recently showed that ensemble methods are useful as a mean to defense against adversarial attacks. However, they only considered implicit ensemble methods which are computationally prohibitive. As our simulations show that explicit ensembles are robust as well, we believe that they are likely to be a useful defense strategy while reducing computational cost. Finally, Theorem 2 suggests that a randomized algorithm can tolerate the non-robustness of some hypotheses to certain samples; this may help to explain Proposition 1 in Kawaguchi et al. (2017): "For any dataset, there exist arbitrarily unstable non-robust algorithms such that has a small generalization gap". We leave this intuition for future work.

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

# Supplementary material

## 7 ADDITIONAL SIMULATIONS

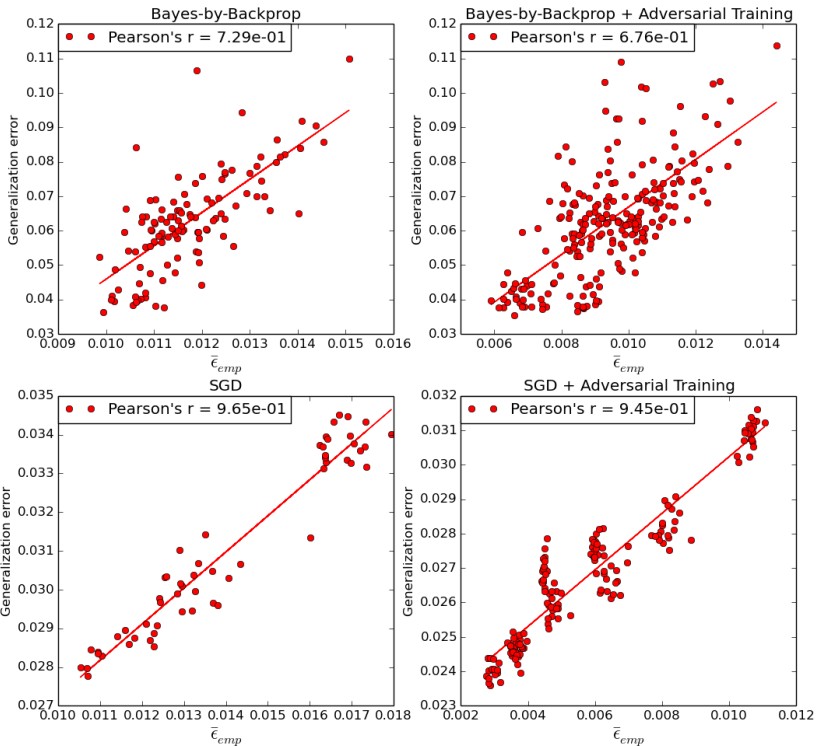

Figure 2: Results for **notMNIST**. Empirical ensemble robustness $\bar{\epsilon}_{\mathrm{emp}}$ (x-axis) vs generalization error (y-axis). Results are given for four different deep learning algorithms.

## 8 UNDERSTANDING DROPOUT VIA ENSEMBLE ROBUSTNESS

In this section, we illustrate how ensemble robustness can well characterize the performance of various training strategies of deep learning. In particular, we take the dropout as a concrete example.

Dropout is a widely used technique for optimizing deep neural network models. We demonstrate that dropout is a random scheme to perturb the algorithm. During dropout, at each step, a random fraction of the units are masked out in a round of parameter updating.

**Assumption 1.** *We assume the randomness of the algorithm $\mathcal{A}$ is parametrized by $\mathbf{r} = (\mathbf{r}_1, \ldots, \mathbf{r}_L) \in \mathcal{R}$ where $\mathbf{r}_l, l = 1, \ldots, L$ are random elements drawn independently.*

For a deep neural network consisting of $L$ layers, the random variable $\mathbf{r}_l$ is the dropout randomness for the $l$-th layer. The next theorem establishes the generalization performance for the neural network with dropout training.

**Theorem 3** (Generalization of Dropout Training). *Consider an L-layer neural network trained by dropout. Let $\mathcal{A}$ be an algorithm with $(K, \bar{\epsilon}(n))$ ensemble robustness. Let $\Delta(\mathcal{H})$ denote the output*

*hypothesis distribution of the randomized algorithm $\mathcal{A}$ on a training set $\mathbf{s}$. Assume there exists a $\beta > 0$ such that,*

$$\sup_{\mathbf{r},\mathbf{t}} \sup_{z \in \mathcal{Z}} |\ell(\mathcal{A}_{\mathbf{s},\mathbf{r}}, z) - \ell(\mathcal{A}_{\mathbf{s},\mathbf{t}}, z)| \leq \beta \leq L^{-3/4},$$

*with $\mathbf{r}$ and $\mathbf{t}$ only differing in one element. Then for any $\delta > 0$, with probability at least $1 - \delta$ with respect to the random draw of the $\mathbf{s}$ and $h \sim \Delta(\mathcal{H})$,*

$$\mathcal{L}(h_{\mathbf{s},\mathbf{r}}) - \ell_{\mathrm{emp}}(h_{\mathbf{s},\mathbf{r}}) \leq \bar{\epsilon}(n) + \sqrt{2\log(1/\delta)/L} + \sqrt{\frac{2K \ln 2 + 2\ln(2/\delta)}{n}}.$$

Theorem 3 also establishes the relation between the depth of a neural network model and the generalization performance. It suggests that when using dropout training, controlling the variance $\beta$ of the empirical performance over different runs is important: when $\beta$ converges at the rate of $L^{-3/4}$, increasing the layer number $L$ will improve the performance of a deep neural network model. However, simply making $L$ larger without controlling $\beta$ does not help. Therefore, in practice, we usually use voting from multiple models to reduce the variance and thus decrease the generalization error (Hinton et al., 2014). Also, when dropout training is applied for more layers in a neural network model, smaller variance of the model performance is preferred. This can be compensated by increasing the size of training examples or ensemble of multiple models.

## 9 TECHNICAL LEMMAS

**Lemma 1.** *For a randomized learning algorithm $\mathcal{A}$ with $(K, \bar{\epsilon}(n))$ uniform ensemble robustness, and loss function $\ell$ such that $0 \leq \ell(h, z) \leq M$, we have,*

$$\mathbb{P}_{\mathbf{s}} \left\{ \mathbb{E}_{\mathcal{A}} |\mathcal{L}(h) - \ell_{\mathrm{emp}}(h)| \leq \bar{\epsilon}(n) + M \sqrt{\frac{2K \ln 2 + 2\ln(1/\delta)}{n}} \right\} \geq 1 - \delta,$$

*where we use $\mathbb{P}_{\mathbf{s}}$ to denote the probability w.r.t. the choice of $\mathbf{s}$, and $|\mathbf{s}| = n$.*

*Proof.* Given a random choice of training set $\mathbf{s}$ with cardinality of $n$, let $N_i$ be the set of index of points of $\mathbf{s}$ that fall into the $C_i$. Note that $(|N_1|, \ldots, |N_K|)$ is an i.i.d. multinomial random variable with parameters $n$ and $(\mu(C_1), \ldots, \mu(C_K))$. The following holds by the Breteganolle-Huber-Carol inequality:

$$\mathbb{P}_{\mathbf{s}} \left\{ \sum_{i=1}^{K} \left| \frac{|N_i|}{n} - \mu(C_i) \right| \geq \lambda \right\} \leq 2^K \exp\left( \frac{-n\lambda^2}{2} \right).$$

We have

$$\mathbb{E}_{\mathcal{A}}|\mathcal{L}(\mathcal{A}_\mathbf{s}) - \ell_{\text{emp}}(\mathcal{A}_\mathbf{s})|$$

$$= \mathbb{E}_{\mathcal{A}} \left| \sum_{i=1}^{K} \mathbb{E}_{z \sim \mu}(\ell(\mathcal{A}_\mathbf{s}, z)|z \in C_i)\mu(C_i) - \frac{1}{n} \sum_{i=1}^{n} \ell(\mathcal{A}_\mathbf{s}, s_i) \right|$$

$$\overset{(a)}{\leq} \mathbb{E}_{\mathcal{A}} \left| \sum_{i=1}^{K} \mathbb{E}_{z \sim \mu}(\ell(\mathcal{A}_\mathbf{s}, z)|z \in C_i)\frac{|N_i|}{n} - \frac{1}{n} \sum_{i=1}^{n} \ell(\mathcal{A}_\mathbf{s}, s_i) \right|$$

$$+ \mathbb{E}_{\mathcal{A}} \left| \sum_{i=1}^{K} \mathbb{E}_{z \sim \mu}(\ell(\mathcal{A}_\mathbf{s}, z)|z \in C_i)\mu(C_i) - \sum_{i=1}^{K} \mathbb{E}_{z \sim \mu}(\ell(\mathcal{A}_\mathbf{s}, z)|z \in C_i)\frac{|N_i|}{n} \right|$$

$$\overset{(b)}{\leq} \sum_{i=1}^{K} \mathbb{E}_{\mathcal{A}} \left| \mathbb{E}_{z \sim \mu}(\ell(\mathcal{A}_\mathbf{s}, z)|z \in C_i)\frac{|N_i|}{n} - \frac{1}{n} \sum_{j \in N_i} \ell(\mathcal{A}_\mathbf{s}, s_j) \right|$$

$$+ \mathbb{E}_{\mathcal{A}} \left| \sum_{i=1}^{K} \mathbb{E}_{z \sim \mu}(\ell(\mathcal{A}_\mathbf{s}, z)|z \in C_i)\mu(C_i) - \sum_{i=1}^{K} \mathbb{E}_{z \sim \mu}(\ell(\mathcal{A}_\mathbf{s}, z)|z \in C_i)\frac{|N_i|}{n} \right|$$

$$\leq \frac{1}{n} \sum_{i=1}^{K} \sum_{j \in N_i} \mathbb{E}_{\mathcal{A}} \left( \max_{z \in C_i} |\ell(\mathcal{A}_\mathbf{s}, s_j) - \ell(\mathcal{A}_\mathbf{s}, z)| \right) + \max_{z \in \mathcal{Z}} |\ell(\mathcal{A}_\mathbf{s}, z)| \sum_{i=1}^{K} \left| \frac{|N_i|}{n} - \mu(C_i) \right| \quad (4)$$

$$\overset{(c)}{\leq} \bar{\epsilon}(n) + M \sum_{i=1}^{K} \left| \frac{|N_i|}{n} - \mu(C_i) \right|$$

$$\overset{(d)}{\leq} \bar{\epsilon}(n) + M \sqrt{\frac{2K \ln 2 + 2 \ln(1/\delta)}{n}} \qquad (5)$$

Here the inequalities $(a)$ and $(b)$ are due to triangle inequality, $(c)$ is from the definition of ensemble robustness and the fact that the loss function is upper bounded by $M$, and $(d)$ holds with a probability greater than $1 - \delta$. $\qquad \square$

**Lemma 2.** *For a randomized learning algorithm $\mathcal{A}$ with $(K, \bar{\epsilon}(n))$ uniform ensemble robustness, and loss function $\ell$ such that $0 \leq \ell(h, z) \leq M$, we have,*

$$\mathbb{E}_\mathbf{s}|\mathcal{L}(h) - \ell_{\text{emp}}(h)|^2 \leq M\bar{\epsilon}(n) + \frac{2M^2}{n}.$$

*Proof.* Let $N_i$ be the set of index of points of $\mathbf{s}$ that fall into the $C_i$. Note that $(|N_1|, \ldots, |N_K|)$ is an i.i.d. multinomial random variable with parameters $n$ and $(\mu(C_1), \ldots, \mu(C_K))$. Then $\mathbb{E}_\mathbf{s}|N_k| =$

$n \cdot \mu(C_k)$ for $k = 1, \ldots, K$.

$$\mathbb{E}_{\mathbf{s}} |\mathcal{L}(h) - \ell_{\mathrm{emp}}(h)|^2$$

$$= \mathbb{E}_{\mathbf{s}} \left| \mathbb{E}_{z \in \mathcal{Z}} \ell(h, z) - \frac{1}{n} \sum_{i=1}^{n} \ell(h, s_i) \right|^2$$

$$= \mathbb{E}_{\mathbf{s}} \left| \sum_{k=1}^{K} \mathbb{E}_{z \in \mathcal{Z}} \ell(h, z | z \in C_k) \mu(C_k) - \frac{1}{n} \sum_{i=1}^{n} \ell(h, s_i) \right|^2$$

$$= \left( \sum_{k=1}^{K} \mathbb{E}_{z \in \mathcal{Z}} \ell(h, z | z \in C_k) \mu(C_k) \right)^2 + \frac{1}{n^2} \mathbb{E}_{\mathbf{s}} \left( \sum_{i=1}^{n} \ell(h, s_i) \right)^2$$

$$\quad - 2 \left( \sum_{k=1}^{K} \mathbb{E}_{z \in \mathcal{Z}} \ell(h, z | z \in C_k) \mu(C_k) \right) \mathbb{E}_{\mathbf{s}} \left( \frac{1}{n} \sum_{i=1}^{n} \ell(h, s_i) \right)$$

$$\leq \left( \sum_{k=1}^{K} \mathbb{E}_{z \in \mathcal{Z}} \ell(h, z | z \in C_k) \mu(C_k) \right) \left| \sum_{k=1}^{K} \mathbb{E}_{z \in \mathcal{Z}} \ell(h, z | z \in C_k) \mu(C_k) - \mathbb{E}_{\mathbf{s}} \frac{1}{n} \sum_{i=1}^{n} \ell(h, s_i) \right|$$

$$\quad + \frac{1}{n^2} \mathbb{E}_{\mathbf{s}} \left( \sum_{i=1}^{n} \ell(h, s_i) \right)^2 - \left( \sum_{k=1}^{K} \mathbb{E}_{z \in \mathcal{Z}} \ell(h, z | z \in C_k) \mu(C_k) \right) \mathbb{E}_{\mathbf{s}} \left( \frac{1}{n} \sum_{i=1}^{n} \ell(h, s_i) \right)$$

$$\leq M \underbrace{\left| \sum_{k=1}^{K} \mathbb{E}_{z \in \mathcal{Z}} \ell(h, z | z \in C_k) \mu(C_k) - \mathbb{E}_{\mathbf{s}} \frac{1}{n} \sum_{i=1}^{n} \ell(h, s_i) \right|}_{H} + \frac{2M^2}{n}$$

We then bound the term $H$ as follows.

$$H = \left| \sum_{k=1}^{K} \mathbb{E}_{z \in \mathcal{Z}} \ell(h, z | z \in C_k) \mathbb{E}_{\mathbf{s}} \frac{N_k}{n} - \mathbb{E}_{\mathbf{s}} \frac{1}{n} \sum_{i=1}^{n} \ell(h, s_i) \right|$$

$$\leq \frac{1}{n} \mathbb{E}_{\mathbf{s}} \left| \sum_{k=1}^{K} \mathbb{E}_{z \in \mathcal{Z}} \ell(h, z | z \in C_k) N_k - \sum_{j \in C_k} \ell(h, s_j) \right|$$

$$\leq \frac{1}{n} \mathbb{E}_{\mathbf{s}} \sum_{k=1}^{K} N_k \max_{s_j \in C_k, z \in C_k} |\ell(h, z) - \ell(h, s_j)|$$

$$= \frac{1}{n} \sum_{k=1}^{K} N_k \mathbb{E}_{\mathbf{s}} \max_{s_j \in C_k, z \in C_k} |\ell(h, z) - \ell(h, s_j)|$$

$$\leq \bar{\epsilon}(n).$$

Then we have,

$$\mathbb{E}_{\mathbf{s}} |\mathcal{L}(h) - \ell_{\mathrm{emp}}(h)|^2 \leq M \bar{\epsilon}(n) + \frac{2M^2}{n}.$$

$\square$

To analyze the generalization performance of deep learning with dropout, following lemma is central.

**Lemma 3** (Bounded difference inequality (McDiarmid, 1989)). *Let $\mathbf{r} = (\mathbf{r}_1, \ldots, \mathbf{r}_L) \in \mathcal{R}$ be $L$ independent random variables ($\mathbf{r}_l$ can be vectors or scalars) with $\mathbf{r}_l \in \{0, 1\}^{m_l}$. Assume that the function $f : \mathcal{R}^L \to \mathbb{R}$ satisfies:*

$$\sup_{\mathbf{r}^{(l)}, \widetilde{\mathbf{r}}^{(l)}} \left| f(\mathbf{r}^{(l)}) - f(\widetilde{\mathbf{r}}^{(l)}) \right| \leq c_l, \forall l = 1, \ldots, L,$$

*whenever* $\mathbf{r}^{(l)}$ *and* $\widetilde{\mathbf{r}}^{(l)}$ *differ only in the l-th element. Here,* $c_l$ *is a nonnegative function of l. Then, for every* $\epsilon > 0$,

$$\mathbb{P}_{\mathbf{r}} \{f(\mathbf{r}_1, \ldots, \mathbf{r}_L) - \mathbb{E}_{\mathbf{r}} f(\mathbf{r}_1, \ldots, \mathbf{r}_L) \geq \epsilon\}$$
$$\leq \exp\left(-2\epsilon^2 / \sum_{l=1}^{L} c_l^2\right).$$

## 10   PROOF OF THEOREM 1

*Proof of Theorem 1.* Now we proceed to prove Theorem 1. Using Chebyshev's inequality, Lemma 2 leads to the following inequality:

$$\Pr_{\mathbf{s}} \{|\mathcal{L}(h) - \ell_{\text{emp}}(h)| \geq \epsilon|h\} \leq \frac{nM\mathbb{E}_{\mathbf{s}} \max_{s \in \mathbf{s}, z \sim s} |\ell(h, s) - \ell(h, z)| + 2M^2}{n\epsilon^2}.$$

By integrating with respect to $h$, we can derive the following bound on the generalization error:

$$\Pr_{\mathbf{s}, \mathcal{A}} \{|\mathcal{L}(h) - \ell_{\text{emp}}(h)| \geq \epsilon\} \leq \frac{nM\mathbb{E}_{\mathcal{A},s} \max_{s \in \mathbf{s}, z \sim s} |\ell(h, s) - \ell(h, z)| + 2M^2}{n\epsilon^2}.$$

This is equivalent to:

$$|\mathcal{L}(h) - \ell_{\text{emp}}(h)| \leq \sqrt{\frac{nM\bar{\epsilon}(n) + 2M^2}{\delta n}}$$

holds with a probability greater than $1 - \delta$. □

## 11   PROOF OF THEOREM 2

*Proof.* To simplify the notations, we use $X(h)$ to denote the random variable $\max_{z \sim s} |\ell(h, s) - \ell(h, z)|$. According to the definition of ensemble robustness, we have $\mathbb{E}_{\mathcal{A}} X(h) \leq \epsilon(n)$. Also, the assumption gives $\text{var}[X(h)] \leq \alpha$. According to Chebyshev's inequality, we have,

$$\mathbb{P}\left\{X(h) \leq \epsilon(n) + \frac{\alpha}{\sqrt{\delta}}\right\} \geq 1 - \delta.$$

Now, we proceed to bound $|\mathcal{L}(h) - \ell_{\text{emp}}(h)|$ for any $h \sim \Delta(\mathcal{H})$ output by $\mathcal{A}_{\mathbf{s}}$.

Following the proof of Lemma 2, we also divide the set $\mathcal{Z}$ into $K$ disjoint set $C_1, \ldots, C_K$ and let $N_i$ be the set of index of points in $\int$ that fall into $C_i$. Then we have,

$$|\mathcal{L}(h) - \ell_{\text{emp}}(h)|$$
$$\leq \frac{1}{n} \sum_{i=1}^{K} \sum_{j \in N_i} \max_{z \in C_i} |\ell(h, s_j) - \ell(h, z)| + \sqrt{\frac{2K \ln 2 + 2 \ln(1/\delta)}{n}}$$
$$\leq \epsilon(n) + \frac{\alpha}{\sqrt{\delta}} + \sqrt{\frac{2K \ln 2 + 2 \ln(1/\delta)}{n}}$$

holds with probability at least $1 - 2\delta$. Let $\delta$ be $2\delta$, we have,

$$|\mathcal{L}(h) - \ell_{\text{emp}}(h)| \leq \epsilon(n) + \frac{\alpha}{\sqrt{2\delta}} + \sqrt{\frac{2K \ln 2 + 2 \ln(1/2\delta)}{n}}$$

holds with probability at least $1 - \delta$. This gives the first inequality in the theorem. The second inequality can be straightforwardly derived from the fact that $\text{var}(X) = \mathbb{E}[X^2] - (\mathbb{E}[X])^2 \leq M\mathbb{E}[X] - (\mathbb{E}[X])^2$. □

## 12  PROOF OF THEOREM 3

*Proof.* Let $R(\mathbf{s}, \mathbf{r}) = \mathcal{L}(\mathcal{A}_{\mathbf{s},\mathbf{r}}) - \ell_{\text{emp}}(\mathcal{A}_{\mathbf{s},\mathbf{r}})$ denote the random variable that we are going to bound. For every $\mathbf{r}, \mathbf{t} \in \mathcal{R}^L$, and $L \in \mathbb{N}$, we have

$$|R(\mathbf{s}, \mathbf{r}) - R(\mathbf{s}, \mathbf{t})|$$

$$= \left| \mathbb{E}_{z \in \mathcal{Z}} \left[ \ell(\mathcal{A}_{\mathbf{s},\mathbf{r}}, z) - \ell(\mathcal{A}_{\mathbf{s},\mathbf{t}}, z) \right] - \frac{1}{n} \sum_{i=1}^{n} \left( \ell(\mathcal{A}_{\mathbf{s},\mathbf{r}}, z_i) - \ell(\mathcal{A}_{\mathbf{s},\mathbf{t}}, z_i) \right) \right|$$

$$\leq \mathbb{E}_{z \in \mathcal{Z}} \left| \ell(\mathcal{A}_{\mathbf{s},\mathbf{r}}, z) - \ell(\mathcal{A}_{\mathbf{s},\mathbf{t}}, z) \right| + \frac{1}{n} \sum_{i=1}^{n} \left| \ell(\mathcal{A}_{\mathbf{s},\mathbf{r}}, z_i) - \ell(\mathcal{A}_{\mathbf{s},\mathbf{t}}, z_i) \right|.$$

According to the definition of $\beta$:

$$\sup_{\mathbf{r},\mathbf{t}} |R(\mathbf{s}, \mathbf{r}) - R(\mathbf{s}, \mathbf{t})| \leq 2\beta,$$

and applying Lemma 3 we obtain (note that $\mathbf{s}$ is independent of $\mathbf{r}$)

$$\mathbb{P}_{\mathbf{r}} \left\{ R(\mathbf{s}, \mathbf{r}) - \mathbb{E}_{\mathbf{r}} R(\mathbf{s}, \mathbf{r}) \geq \epsilon | \mathbf{s} \right\} \leq \exp\left( \frac{-\epsilon^2}{2L\beta^2} \right).$$

We also have

$$\mathbb{E}_{\mathbf{s}} \mathbb{P}_{\mathbf{r}} \left\{ R(\mathbf{s}, \mathbf{r}) - \mathbb{E}_{\mathbf{r}} R(\mathbf{s}, \mathbf{r}) \geq \epsilon \right\} = \mathbb{E}_{\mathbf{s}} \mathbb{P}_{\mathbf{r}} \left\{ R(\mathbf{s}, \mathbf{r}) - \mathbb{E}_{\mathbf{r}} R(\mathbf{s}, \mathbf{r}) \geq \epsilon | \mathbf{s} \right\} \leq \exp\left( \frac{-\epsilon^2}{2L\beta^2} \right).$$

Setting the r.h.s. equal to $\delta$ and writing $\epsilon$ as a function of $\delta$, we have that with probability at least $1 - \delta$ w.r.t. the random sampling of $\mathbf{s}$ and $\mathbf{r}$:

$$R(\mathbf{s}, \mathbf{r}) - \mathbb{E}_{\mathbf{r}} R(\mathbf{s}, \mathbf{r}) \leq \beta\sqrt{2L\log(1/\delta)}.$$

Then according to Lemma 1:

$$\mathbb{E}_{\mathbf{r}} R(\mathbf{s}, \mathbf{r}) \leq \bar{\epsilon}(n) + \sqrt{\frac{2K\ln 2 + 2\ln(1/\delta)}{n}}$$

holds with probability greater than $1 - \delta$. Observe that the above two inequalities hold simultaneously with probability at least $1 - 2\delta$. Combining those inequalities and setting $\delta = \delta/2$ gives

$$R(\mathbf{s}, \mathbf{r}) \leq \beta\sqrt{2L\log(1/\delta)} + \bar{\epsilon}(n) + \sqrt{\frac{2K\ln 2 + 2\ln(2/\delta)}{n}}.$$

$\square$

