# OpenReview forum: "Ensemble Robustness and Generalization of Stochastic Deep Learning Algorithms"
_ICLR.cc/2018/Conference — Invite to Workshop Track_

### Official Review · AnonReviewer1 · 2017-11-24
**Paper that provides a novel theoretical framework for stochastic learning of Deep Networks, the proposed framework is an extension of an existing framework and the contribution is a bit limited in its present form.**

**Rating:** 4
**Confidence:** 3

**Review:**

Summary:
This paper presents an adaptation of the algorithmic robustness of Xu&Mannor'12 to a notion robustness of ensemble of hypothesis allowing the authors to study generalization ability of stochastic learning algorithms for Deep Learning Networks.
Generalization can be established as long as the sensitiveness of the learning algorithm to adversarial perturbations is bounded.
The paper presents learning bounds and an experimental showing correlation between empirical ensemble robustness and generalization error.

Quality:
Globally correct

Clarity:
Paper clear

Originality:
Limited with respect to the original definition of algorithmic robustness

Significance:
The paper provides a new theoretical analysis for stochastic learning of Deep Networks but the contribution is limited in its present form.


Pros:
-New theoretical study for DL algorithms
-Focus on adversarial learning
Cons
-I find the contribution a bit limited
-Some aspects have to be precised/more argumented
-Experimental study could have been more complete


Comments:
---------


*About the proposed framework.
The idea of taking a max over instances of partition C_i (Def 3) already appeared in the proof of results of Xu&Mannor, and the originality of the contribution is essentially to add an expectation over the result of the algorithm.


In Xu&Mannor paper, there is a notion of weak robustness that is proved to be necessary and sufficient to generalize. The contribution of the authors would be stronger if they can discuss an equivalent notion in their context.

The partition considered by the framework is never discussed nor taken into account, while this is an important aspect of the analysis. In particular, there is a tradeoff between \epsilon(s) and K:  using a very fine tiling it is always possible to have a very small \epsilon(s) at the price of a very large K (if you think of a covering number, K can be exponential in the size of the tiling and hard to calculate).
In the context of adversarial examples, this is actually important because it can be very likely that the adversarial example can belong to a partition set different from the set the original example belong to.
In this context, I am not sure to understand the validity of the framework because we can then compare 2 instances of different set which is outside of the framework.
So I wonder if the way the adervarial examples are generates should be taken into account for the definition of the partition.
Additionnally, the result is given in the contect of IID data, and with a multinomial distribution according to the partition set - adversarial generation can violate this IID assumption.

In the experimental setup, the partition set is not explained and we have no guarantee to compare instances of the same set. Nothing is said about $r$ and its impact on the results. This is a clear weak aspect of the experimental analysis
In the experimental setup, as far as I understand the setup, I find the term "generalization error" a bit abusive since it is actually the error on the test set.
Using cross validation or considering multiple training/test sets would be more appropriate.


In the proof of Lemma 2, I am not sure to understand where the term 1/n comes from in the term 2M^2/2 (before "We then bound the term H as follows")

---

> ### Author Response · Authors · 2017-12-24
> **Reply**
>
> We thank the reviewer for pointing these issues out and agree that they were not explained well.  We have revised the paper to explain the data partitioning principles better and address here the main points the reviewer raises.
>
> Regarding partition for sets:
> Generally, there is a trade-off between epsilon(s) and, K, the larger K is the smaller \epsilon(s) due to the finer tiling as the reviewer suggested. This tradeoff is also evident in the bound of Theorem 1, where the right-hand side increases with K and \epsilon(s) so there is a minimum point (see Corollaries 4&5 in Xu&Mannor 2012 for choosing the minimal K).
>
> However, in the context of Deep Neural Networks, we chose k=n (training data size), to be an implicit partition such that each set contains a small R2 ball around each training example, without specifying the partition explicitly. We then approximate the loss in this partition using the adversarial example, i.e., approximating the maximal loss in the partition using the adversarial example. While this approximation is loose, we show that empirically, it is correlated with generalization. Under this partition, there is no violation of the IID assumption for general stochastic algorithms, but it is violated in the case of adversarial training as the reviewer suggested. However, simulations suggest that correlation exists for both.
>
>
> Regarding weak robustness: We are more interested in the standard generalizability and found weak robustness to be out of the scope of this work. We do believe however that similar bound can be derived for weak robustness of randomized algorithms using the same techniques we used in this work.

---

### Official Review · AnonReviewer2 · 2017-11-26
**Paper proposes to study generalization ability via new notion of stability. Improtant problem and a beginning of interesting idea but paper is not ready for a publication since results are not sufficiently strong.**

**Rating:** 4
**Confidence:** 5

**Review:**

This paper proposes a study of the generalization ability of deep learning algorithms using an extension of notion of stability called ensemble robustness. It requires that algorithm is stable on average with respect to randomness of the algorithm. The paper then gives bounds on the generalization error of a randomized algorithm in terms of stability parameter and provides empirical study attempting to connect theory with practice.

While I believe that paper is trying to tackle an important problem and maybe on the right path to find notions that are responsible for generalization in NNs, I believe that contributions in this work are not sufficiently strong for acceptance.

Firstly, it should be noted that the notion of generalization considered in this work is significantly weaker than standard notions of generalization in learning theory since (a) results are not high probability results (b) the bounds are with respect to randomness of both sample and sample (which gives extra slack).

Stabiltiy parameter epsilon_bar(n) is not studied anywhere. How does it scale with sample size n for standard algorithms? How do we know it does not make bounds vacuous?

It is only allude it to that NN learning algorithms may poses ensemble robustness. It is not clear and not shown anywhere that they do. Indeed, simulations demonstrate that this could be the case but this still presents a significant gap between theory and practice (just like any other analysis that paper criticizes in intro).

Minor:

1. After Theorem 2: "... can substantially improve ..." not sure if improvement is substantial since it is still not a high probability bound.

2. In intro, "Thus statistical learning theory ... struggle to explain generalization ...". Note that the work of Zhang et al does not establish that learning theory struggle to explain generalization ability of NNs since results in that paper do not study margin bounds. To this end refer to some recent work Bartlett et al, Cortes et al., Neyshabur et al.

3. Typos in def. 3. missing z in "If s \in C_i...". No bar on epsilon.

---

> ### Author Response · Authors · 2017-12-24
> **Reply**
>
> We thank the reviewer for his feedback.
>
> Regarding epsilon_bar(n): While the study of epsion_bar(n) is hard in the context of general algorithms and deep networks, it can be done for simpler learning algorithms. For example, for linear SVM, \epsilon_bar(n) will be relevant to the covering number (robustness and regularization of support vector machines, Xu et. al. 09).
>
> Regarding the robustness of NNs:
> We agree it is hard to show explicitly that NNs are robust. This is exactly the goal of this paper, trying to bridge the gap between theory and practice. We want to emphasize that the goal of this paper is not to criticize other methods, but to provide a different perspective.
>
> Regarding high probability bounds: Can the reviewer explain what he means by these two comments? (a) Our theorems are given in the PAC epsilon/delta formulation which is, in fact, a high probability bound. (b) We do not understand what the reviewer means by the randomness of both sample and sample.
>
> All minor comments that the reviewers mentioned were fixed in the pdf.

---

> > ### Comment · AnonReviewer2 · 2018-01-05
> > **reply**
> >
> > "Regarding epsilon_bar(n): While the study of epsion_bar(n) is hard in the context of general algorithms and deep networks, it can be done for simpler learning algorithms. For example, for linear SVM, \epsilon_bar(n) will be relevant to the covering number (robustness and regularization of support vector machines, Xu et. al. 09)."
> >
> > --> But how do we now the bound is not trivial in case of deep nets?
> >
> >
> > "Regarding high probability bounds: Can the reviewer explain what he means by these two comments? (a) Our theorems are given in the PAC epsilon/delta formulation which is, in fact, a high probability bound. (b) We do not understand what the reviewer means by the randomness of both sample and sample. "
> >
> > --> Standard results in ML are logarithmic in 1/delta, these results are only linear in 1/delta which is a very weak result.
> >
> > --> I meant randomness of sample and algorithms.

---

> ### Public Comment · (anonymous) · 2018-01-04
> **Bounds do in fact hold with high probability**
>
> With all due respect, I feel that this review is mistaken about the bounds not holding with high probability. Theorems 1 & 2 clearly state that the bounds hold "with probability at least $1 − \delta$ with respect to the random draw of the s and h."
>
> That said, the bounds are _linear_ in $1/\delta$, which is not ideal; it would be stronger if they were logarithmic in $1/\delta$. (Note: Theorem 2 has a term that is linear in $1/\delta$, which becomes the dominating term.)

---

> > ### Comment · AnonReviewer2 · 2018-01-05
> > **Bounds do NOT hold with high probability**
> >
> > Logarithmic dependence on 1/delta is what is understood under "high probability". This is standard in ML theory see for instance definition of PAC learning.

---

### Official Review · AnonReviewer3 · 2017-12-01
**Clear accept**

**Rating:** 8
**Confidence:** 4

**Review:**

The paper studied the generalization ability of learning algorithms from the robustness viewpoint in a deep learning context. To achieve this goal, the authors extended the notion of the (K, \epsilon)- robustness proposed in Xu and Mannor, 2012 and introduced the ensemble robustness.

Pros:

1, The problem studied in this paper is interesting. Both robustness and generalization are important properties of learning algorithms. It is good to see that the authors made some efforts towards this direction.
2, The paper is well shaped and is easy to follow. The analysis conducted in this paper is sound. Numerical experiments are also convincing.
3, The extended notion "ensemble robustness" is shown to be very useful in studying the generalization properties of several deep learning algorithms.

Cons:

1,  The terminology "ensemble" seems odd to me, and seems not to be informative enough.
2,  Given that the stability is considered as a weak notion of robustness, and the fact that the stability of a learning algorithm and its relations to the generalization property have been well studied, in my view, it is quite necessary to mention the relation of the present study with stability arguments.
3, After Definition 3, the author stated that ensemble robustness is a weak notion of robustness proposed in Xu and Manner, 2012. It is better to present an example here immediately to illustrate.

---

> ### Author Response · Authors · 2017-12-24
> **Reply**
>
> We thank the reviewer for his feedback.
>
> Regarding the 3 cons the reviewer mentioned:
>
> 1. We agree that a better terminology may be found, at the moment we decided to stick to the original one.
> 2. We have addressed point two in the forum and in the new version of the pdf (related work Section).
> 3. Good point. We moved this discussion to after theorem two and revisited the discussion after theorem 2 to explain this issue better.

---

### Public Comment · (anonymous) · 2017-11-29
**Please discuss relationship to randomized stability-based bounds**

Ensemble robustness is conceptually very similar to randomized algorithm stability. The latter concept has been thoroughly analyzed by Elisseeff et al. (JMLR, 2005), who derived a number of generalization bounds for randomized algorithms based on different notions of stability (uniform, hypothesis, pointwise hypothesis). Given the similarity between robustness and stability, it seems to me that the submitted paper should discuss the connections to Elisseeff et al.'s work (which is not cited) and compare the bounds in both.

---

> ### Author Response · Authors · 2017-11-30
> **Reply to: Please discuss relationship to randomized stability-based bounds**
>
> Thank you for your comment. Stability and robustness are two examples of desired properties of a learning algorithm that can also guarantee generalization under some conditions.
>
> A stable algorithm produces an output hypothesis that is stable to small changes in the data set, i.e., if a training example is replaced with another example from the same distribution, the training error will not change much. Elisseeff et al. (JMLR, 2005), indeed showed that algorithm that fulfills this requirement generalize well.
>
> Robustness, on the other hand, is a different property of learning algorithms. A Robust algorithm produces a hypothesis that is robust to bounded perturbations of the entire data set, as we explain in more detail in our paper. Robustness and Stability are different properties, to see that observe that robustness is a global property while stability is local, and that robustness concerns properties of a single hypothesis, while stability concerns two (one for the original data set and one for the modified one).
>
> We emphasize that a learning algorithm may be both stable and robust, e.g., SVM, "Robustness and Regularization of Support Vector Machines," Huan Xu, Constantine Caramanis, Shie Mannor 2009). However, there also exist algorithms that are robust but not stable, e.g., Lasso Regression, "Robust Regression and Lasso," Huan Xu, Constantine Caramanis, Shie Mannor 2008).
>
> We will further expand the discussion on these issues in a future revision of the paper.

---

> > ### Public Comment · (anonymous) · 2018-01-05
> > **Explanation could use some polishing and further discussion**
> >
> > "Robustness and Stability are different properties, to see that observe that robustness is a global property while stability is local, and that robustness concerns properties of a single hypothesis, while stability concerns two (one for the original data set and one for the modified one). "
> >
> > I don't think "global vs local" is the best way to distinguish robustness from stability. Both robustness and stability bound the affect of local perturbations. The key difference, IMO, is that robustness deals with perturbations of the test example, whereas stability deals with perturbations of a single training example. Robustness also constrains the test example perturbations to be within a certain partition of the instance space, whereas stability allows the perturbations to range over the entire instance space.
> >
> > For algorithms that are both robust and stable, which analysis yields better bounds? Since the paper is concerned with deep learning, consider the stability results in Hardt et al. (2016) or Kuzborskij & Lampert (2017) for learning with non-convex objectives. If one were to combine these results with Elisseeff et al.'s generalization bounds, would the resulting bounds be better or worse than the ones in this paper? I'm just saying that more comparison to related work would make the paper stronger.

---

### Author Response · Authors · 2017-12-24
**General comments for the reviewers**

We thank the reviewers for their constructive feedback, which we found very helpful to improve the quality of this work. For each of the reviewers, a personal response is posted in the forum. Also, a new revision of the paper is available following the reviewer remarks. For the reviewer convenience, additions/corrections are marked in a red color in the text to distinguish new text from old one.

Here, we would like to emphasize the contributions of this paper and its importance to the ICLR community as we see it. This paper revisits the robustness=generalization theory (Xu & Mannor, 2012) in the context of deep networks. We introduce new theorems that deal with stochastic algorithms (the most deployed ones) and provide a complimentary empirical study on the connection between robustness and generalization of Deep Neural Networks. We provide for the first time, an empirical study on the global (as we define it) robustness of the Deep Neural Networks, and its connections to generalization and adversarial examples, which has been puzzling the Deep Learning community lately. Moreover, we have shown that taking an expectation over robustness indeed improves the correlation between robustness and generalization, which we later demonstrate how to evaluate efficiently through Bayesian networks. Finally, we believe that the study of different approaches for generalization of Deep Neural Nets is of high importance, and we believe that this work makes an interesting step in this direction.

For each of the reviewers, a personal response is posted in the forum. Also, a new revision of the paper is available following the reviewer remarks. For the reviewer convenience, additions/corrections are marked in a red color in the text to distinguish new text from old one. Main modifications:

·       Intro: a few clarifications about our claims and fixing of citations following R2 comments.
·       Intro: better discussion on adversarial training and Parseval networks.
·       Related work: discussion on stability following comment in the forum + R1.
·       Better explanation of partitions sets, experimental considerations of them – Sections 4, and 5. (R3)

---

### Decision · Program_Chairs · 2018-01-29
**ICLR 2018 Conference Acceptance Decision**

**Decision:**

Invite to Workshop Track

**Comment:**

The paper proposes a new way to understand why neural networks generalize well. They introduce the concept of ensemble robustness and try to explain DNN generalization based on this concept. The reviewers feel the paper is a bit premature for publication in a top conference although this new way of explaining generalization is quite interesting.